# Autobidder's Dilemma: Why More Sophisticated Autobidders Lead to Worse Auction Efficiency

**Yuan Deng**
Google Research
dengyuan@google.com

**Jieming Mao**
Google Research
maojm@google.com

**Vahab Mirrokni**
Google Research
mirrokni@google.com

**Hanrui Zhang**
Chinese University of Hong Kong
hanrui@cse.cuhk.edu.hk

**Song Zuo**
Google Research
szuo@google.com

## Abstract

The recent increasing adoption of autobidding has inspired the growing interest in analyzing the performance of classic mechanism with value-maximizing auto-bidders both theoretically and empirically. It is known that optimal welfare can be obtained in first-price auctions if autobidders are restricted to uniform bid-scaling and the price of anarchy is 2 when non-uniform bid-scaling strategies are allowed.

In this paper, we provide a fine-grained price of anarchy analysis for non-uniform bid-scaling strategies in first-price auctions, demonstrating the reason why more powerful (individual) non-uniform bid-scaling strategies may lead to worse (aggregated) performance in social welfare. Our theoretical results match recent empirical findings that a higher level of non-uniform bid-scaling leads to lower welfare performance in first-price auctions.

## 1   Introduction

The online advertising market has witnessed increasing adoption of autobidding in recent years, which shifts the bidding behavior model of advertisers from the classic utility-maximizing bidding to value-maximizing bidding [Aggarwal et al., 2019, Balseiro et al., 2021b]. Unlike the classic utility maximizers who maximize their quasi-linear utility given by the difference between value and payment, value maximizers maximize the total value subject to a return-on-spend (RoS) constraint [Balseiro et al., 2021b].

The shift in the bidding behavior model has motivated a growing body of literature on auction design with value-maximizing autobidders. Notably, it has been shown that the price of anarchy (PoA) in second-price auctions [Aggarwal et al., 2019] and first-price auctions [Deng et al., 2022, Liaw et al., 2023] are both 2. However, (1) these PoA results measure the welfare performance in the worst-case scenario (i.e., the worst-case equilibrium of the worst-case instance); moreover, (2) these PoA results assume the bidding agents can find the optimal bidding strategies in response to other bidders so that their bidding profile forms an equilibrium, while computing optimal bidding strategies and/or finding equilibria could be computationally infeasible [Aggarwal et al., 2023, Chen et al., 2021, Li and Tang, 2024, Paes Leme et al., 2024]. On the other hand, Balseiro et al. [2021a] demonstrate that optimal welfare can be obtained in first-price auctions if autobidders are restricted to the simple uniform bid-scaling strategy (i.e., always bid $\theta v$ when the bidder's value is $v$ with a universal bid multiplier $\theta$). But, uniform bid-scaling is not always an optimal strategy for autobidders in first-price auctions as non-uniform bidding scaling, in which autobidder may apply different bid multipliers in different auctions, may result in better bidding performance.

38th Conference on Neural Information Processing Systems (NeurIPS 2024).

To circumvent the above mentioned limitations of PoA results and bridge the gap between uniform bid-scaling and non-uniform bid-scaling, Deng et al. [2024] propose a hierarchical structure over ad auction instances in which the auctions are partitioned to different categories following a multi-layer laminar structure. Autobidders are required to adopt the same bid multipliers for auctions within the same slice but can use different bid multipliers across different slices. Such a multi-layer structure enables the comparison between non-uniform bid-scaling strategies of different degrees of freedom and the empirical results from [Deng et al., 2024] show that a higher level of non-uniform bid-scaling may lead to lower aggregated welfare performance in first-price auctions. In this paper, we take a theoretical approach to non-uniform bid-scaling in first-price marketplaces, aiming to (1) provide a formal understanding of the phenomenon observed by Deng et al. [2024], and (2) establish principles that can guide the design of marketplaces where non-uniform bid-scaling is involved.

## 1.1 Our Results

Our main technical result is a fine-grained analysis of the price of anarchy of automated first-price marketplaces when autobidders are capable of non-uniform bidding. We adopt the high-level model introduced by Deng et al. [2024]: Roughly speaking, the entire market is divided into a number of slices, and autobidders may choose one bid multiplier independently for each slice. Here, intuitively, the granularity of the slices measures autobidders' capability of non-uniform bidding. Through a refined analysis, we present a parameterized price of anarchy bound, which connects the efficiency of an automated marketplace to the power of autobidders measured by their capability of non-uniform bidding, as well as the "balancedness" of slices. Here, the balancedness of a slice roughly measures the smallest "market share" across all bidders. Qualitatively, our results suggest:

- *First-price markets are more efficient when autobidders are less powerful.* This is reminiscent of the prisoner's dilemma, where both prisoners end up in a strictly worse situation, when each of them chooses an action that is superior to the alternative, regardless of the other prisoner's choice. In automated marketplaces, fixing others' bids, a more powerful autobidder always achieves a (weakly) better payoff. And yet, the interplay of such "better" autobidders could lead to worse auction efficiency. This provides a theoretical explanation for the phenomenon observed by Deng et al. [2024].

- *First-price markets are more efficient with more balanced slices.* In particular, when the granularity of non-uniform bidding is fixed, one can improve the efficiency of the market by creating slices that are more balanced. Intuitively, this introduces more intense competition within each slice, resulting in better auction efficiency. Such an insight can also be applied to marketplaces with the multi-channel setting [Deng et al., 2023a, Susan et al., 2023], where advertisers can procure ad impressions simultaneously on multiple channels with different bidding strategies. In the multi-channel setting, more balanced channels could lead to better auction efficiency across all channels.

## 1.2 Further Related Work

The price of anarchy in first-price auctions with quasi-linear utility maximizers has been extensively studied [Roughgarden et al., 2017]. For Bayesian settings, the price of anarchy is at least $1/2$ for subadditive valuations [Feldman et al., 2013] and the price of anarchy is at least $1 - 1/e$ for submodular valuations [Syrgkanis and Tardos, 2013]. When values are independently distributed, Hoy et al. [2018] improved the price of anarchy to $\approx 0.743$. Recently, Jin and Lu [2022] resolve the long-standing open problem and show that the price of anarchy in first-price auctions with independently distributed values is exactly $1 - 1/e^2$.

Our work is also closely related to the recent growing body of research [Aggarwal et al., 2019, Deng et al., 2021, Balseiro et al., 2021a, Mehta, 2022, Deng et al., 2023b, Liaw et al., 2024] which study the price of anarchy (PoA) with value-maximizing bidders in several classic auction mechanisms like second-price auctions, (randomized) first-price auctions, and generalized second-price auctions. In particular, it has been shown the price of anarchy of first-price auctions is exactly $1/2$ [Liaw et al., 2023, Deng et al., 2022]. In the multi-channel setting, Deng et al. [2023a] study the problem of multi-channel bidding where an advertiser aims to maximize their total value across and analyze the effectiveness of levers of return-on-spend and budget. Susan et al. [2023] develop multi-channel bidding algorithms when channels adopt auction rules that may or may not be incentive-compatible

under the presence of budget constraints. More recently, Feng et al. [2023] investigate the PoA of running first-price auctions with strategic budget-constrained autobidders, in which advertisers may manipulate their reported budget constraints for the autobidders. For a more comprehensive overview of auctions and autobidding, see, e.g., [Aggarwal et al., 2024].

## 2 Preliminaries

**The multi-auction model.** Following prior works [Aggarwal et al., 2019, Balseiro et al., 2021b], we consider the following model where multiple bidders participate in multiple auctions simultaneously. There are $n$ bidders (generally indexed by $i$) and $m$ auctions (generally indexed by $j$). In each auction $j$, each bidder $i$ has a value $v_{i,j}$, and places a bid $b_{i,j}$. In this paper, we assume these bids are placed through autobidders using non-uniform bidding, in which the bids are subject to certain restrictions to be discussed below. For brevity, we let $\boldsymbol{v}_i = (v_{i,1}, \ldots, v_{i,m})$, $\boldsymbol{v} = (\boldsymbol{v}_1, \ldots, \boldsymbol{v}_n)$, $\boldsymbol{v}_{-i} = (\boldsymbol{v}_1, \ldots, \boldsymbol{v}_{i-1}, \boldsymbol{v}_{i+1}, \ldots, \boldsymbol{v}_n)$, $\boldsymbol{v}_j = (v_{1,j}, \ldots, v_{n,j})$, and $\boldsymbol{v}_{-i,j} = (v_{1,j}, \ldots, v_{i-1,j}, v_{i+1,j}, \ldots, v_{n,j})$. We use $\boldsymbol{b}$ in a similar way.

Each bidder $i$'s allocation $x_{i,j}$ and payment $p_{i,j}$ in each auction $j$ is determined by the auction rule and all bidders' bids $\boldsymbol{b}_j$ in auction $j$. We focus on first-price auctions: In each auction $j$, the bidder with the highest bid[1] wins, receives the whole item and pays the bid. All other bidders receive nothing and pay 0. Formally, for each auction $j$, let $i^*(j) = \mathrm{argmax}_i\, b_{i,j}$. The allocation $x_{i,j}$ and payment $p_{i,j}$ of each bidder $i$ is given by

$$x_{i,j} = \begin{cases} 1, & \text{if } i = i^*(j) \\ 0, & \text{otherwise} \end{cases}, \qquad \text{and} \qquad p_{i,j} = \begin{cases} b_{i,j}, & \text{if } i = i^*(j) \\ 0, & \text{otherwise} \end{cases}.$$

We will omit the dependency of $x_{i,j}$ and $p_{i,j}$ on $\boldsymbol{b}$ for simplicity. For each auction $j$, we also identify a "rightful winner" $\mathrm{rw}(j) = \mathrm{argmax}_{i \in [n]}\, v_{i,j}$, who has the highest value and therefore should win in auction $j$ in the socially optimal allocation.[2]

**Slices and partially non-uniform bidding.** We capture the power of autobidders beyond uniform bidding with a fine-grained slice-based model [Deng et al., 2024]. Intuitively, we assume that the $m$ auctions are partitioned into slices. Within each slice, the autobidder must bid uniformly, while across slices, it can use different bidding multipliers. Formally, there are $s$ slices (generally indexed by $k$), which together form a partition of the $m$ auctions. Each slice $k$ is described by the set $S_k \subseteq [m]$ of (indices of) auctions it contains. We always have $S_k \cap S_{k'} = \emptyset$ for all $k \neq k'$, and $\bigcup_k S_k = [m]$. For each auction $j \in S_k$, we let $\mathrm{slice}(j) = k$. The bids $b_i$ of each bidder $i$ is generated in the following way: For each slice $k$, the autobidder chooses a bid multiplier $\theta_{i,k}$ such that for each auction $j \in S_k$, $b_{i,j} = v_{i,j} \cdot \theta_{i,k}$. We use $\boldsymbol{\theta}$ in a similar way to $\boldsymbol{v}$ and $\boldsymbol{b}$.

**Constrained value maximizers.** We adopt the value maximization model prevalent in the autobidding literature [Aggarwal et al., 2019, Balseiro et al., 2021b]. The objective of a value maximizer is to maximize the total value that they win in all auctions, subject to the constraint that the overall return on spend (RoS) is larger than a given target, by choosing the optimal bid multipliers for all $s$ slices conditioned on other bidders' bids. In addition, we make the following regularity assumption about bidders' strategies: On each slice $k$, each bidder $i$ never underbids (i.e., never chooses a multiplier $\theta_{i,k} < 1$) unless $i$ wins in all auctions $j \in S_k$ where $i$ is the rightful winner (i.e., where $\mathrm{rw}(j) = i$). The assumption rules out brittle and pathological equilibrium behavior, which allows us to focus on the relation between the power of autobidders and auction efficiency. Intuitively, if a bidder $i$ is forced to underbid on a slice $k$, then $i$ must be overbidding on some other slices (because the overall RoS constraint must be binding — see below for the formal definition). In such cases, the assumption says that autobidders are conservative, in the sense that bidder $i$ would try to secure $i$'s share in the socially optimal allocation on each slice first, before bidding more aggressively and trying to acquire other bidders' share on other slices. We also remark that such regularity assumptions are used in prior work (e.g., [Deng et al., 2021]) to derive meaningful efficiency bounds for other auction formats.

---

[1]We assume ties are broken arbitrarily and consistently throughout the paper.
[2]Ties are broken in the same way as in winner determination.

Formally, let $\mathsf{val}_{i,j} = x_{i,j} \cdot v_{i,j}$ be $i$'s value in auction $j$. Let

$$\mathsf{val}_i = \sum_{j \in [m]} \mathsf{val}_{i,j} \quad \text{and} \quad p_i = \sum_{j \in [m]} p_{i,j}.$$

Moreover, for each bidder $i$ and slice $k$, fixing $\boldsymbol{\theta}_{-i,k}$, let $\underline{\theta_{i,k}}(\boldsymbol{\theta}_{-i,k}) = \min\{1, \underline{\theta}\}$, where

$$\underline{\theta} = \min\{\theta_{i,k} \mid x_{i,j} = 1, \forall j \in S_k \text{ where } \mathsf{rw}(j) = i\}.$$

Let

$$\Theta_i(\boldsymbol{\theta}_{-i}) = [\underline{\theta_{i,1}}(\boldsymbol{\theta}_{-i,1}), \infty) \times \cdots \times [\underline{\theta_{i,s}}(\boldsymbol{\theta}_{-i,s}), \infty).$$

Then, fixing $\boldsymbol{\theta}_{-i}$ (and therefore $\boldsymbol{b}_{-i}$), each bidder $i$ solves the following optimization problem:

$$\max_{\boldsymbol{\theta}_i \in \Theta_i(\boldsymbol{\theta}_{-i})} \quad \mathsf{val}_i$$
$$\text{s.t.} \quad \mathsf{val}_i \geq p_i.$$

We say $\boldsymbol{\theta}_i \in \mathsf{BR}_i(\boldsymbol{\theta}_{-i})$ if $\boldsymbol{\theta}_i$ achieves the maximum of the above optimization problem given $\boldsymbol{\theta}_{-i}$. Note that here, we are assuming that bidder $i$'s target RoS is 1. This is without loss of generality, because scaling each bidder's values by their target RoS preserves the auction outcomes and the liquid welfare, the latter being the standard measure of efficiency with constrained value maximizers.

**Equilibria and price of anarchy.** We consider stable auction outcomes based on Nash equilibria, where all bidders are best responding to each other. Formally, we say a strategy profile $\boldsymbol{\theta}$ is an equilibrium, iff for each bidder $i$, $\boldsymbol{\theta}_i \in \mathsf{BR}_i(\boldsymbol{\theta}_{-i})$. We measure efficiency by considering the standard notion of the Price of Anarchy (PoA), which is the ratio between the liquid welfare at the worst equilibrium and the optimal liquid welfare. Formally, the PoA of a particular instance $(n, m, s, \{S_k\}, \boldsymbol{v})$ in the multi-auction model is defined as follows:

$$\mathsf{PoA}(n, m, s, \{S_k\}, \boldsymbol{v}) = \inf_{\boldsymbol{\theta} \text{ is an equilibrium}} \frac{\sum_{i \in [n]} \mathsf{val}_i}{\sum_{j \in [m]} \max_{i \in [n]} v_{i,j}}.$$

Note that the above definition of PoA is with respect to a single market instance. In later sections, we will adopt a specific parametrization of non-uniformity, and consider the worst-case PoA for any fixed non-uniformity parameters. This allows us to clearly depict the relation between non-uniformity and efficiency of auction outcomes. We also remark that this approach refines the standard PoA analysis, which normally establishes a single worst-case PoA bound over all instances.

## 3 Fine-Grained PoA of First-Price Auctions with Non-Uniform autobidders

In this section, we present and prove the main result of this paper: a fine-grained PoA bound capturing the effect of non-uniform bidding on auction efficiency. As we will discuss later, our result implies that more sophisticated (i.e., more non-uniform) autobidders generally lead to worse auction efficiency.

### 3.1 The Balancedness Parametrization

We first introduce our parametrization of non-uniformity. Fix a market instance $(n, m, s, \{S_k\}, \boldsymbol{v})$.

**Definition 1** (Market share). The market share of a bidder $i$ in a set of auctions $S \subseteq [m]$ is $i$'s contribution to the optimal liquid welfare in $S$, as a fraction of the optimal liquid welfare. Formally,

$$\mathsf{share}_{i,S} = \frac{\sum_{j \in S : \mathsf{rw}(j) = i} v_{i,j}}{\sum_{j \in [m]} \max_{i' \in [n]} v_{i',j}}.$$

**Definition 2** (Balancedness). The balancedness of a set of auctions $S \subseteq [m]$ is the minimum market share of all bidders in $S$, as a fraction of the sum of all bidders' market shares in $S$ (i.e., the contribution of $S$ to the optimal liquid welfare). Formally,

$$\mathsf{bal}_S = \frac{\min_{i \in [n]} \mathsf{share}_{i,S}}{\sum_{i \in [n]} \mathsf{share}_{i,S}}.$$

Without loss of generality, we assume (unless otherwise specified) that slices are numbered such that $\mathsf{bal}_k$ is weakly decreasing in $k$, i.e., for each $k \in [s-1]$, $\mathsf{bal}_{k+1} \leq \mathsf{bal}_k$. Higher balancedness generally means that the contribution to the optimal welfare is more equally distributed among bidders. We will pay special attention to the market share in, and the balancedness of, a whole slice, where $S = S_k$ for some slice $k$. Abusing notation, we let $\mathsf{share}_{i,k} = \mathsf{share}_{i,S_k}$ for each bidder $i$ and slice $k$, and $\mathsf{share}_k = \sum_{i \in [n]} \mathsf{share}_{i,k}$ and $\mathsf{bal}_k = \mathsf{bal}_{S_k}$ for each slice $k$.

We discuss the intuition behind these definitions below. Generally speaking, inefficiency in first-price auctions with value maximizers often manifests in the following way:

- A bidder wins in a large fraction of auctions where that bidder is the rightful winner, without much competition. This means the bidder wins by bidding far below their true value in these auctions, which gives the bidder a significant amount of buyer surplus.
- While the bidder (being a value maximizer) does not intrinsically care about buyer surplus, it allows the bidder to bid more aggressively (i.e., far above their true value) in other auctions, and win extra auctions in which they are not the rightful winners, which makes the auction outcome less efficient.

For autobidders using slice-based non-uniform bidding strategies, since each bidder must bid uniformly (i.e., equally aggressively) within each slice, intuitively, more balanced slices make it harder for the above phenomenon to happen. In particular, in the first bullet point, on a balanced slice, all bidders are bidding seriously because they want to defend their market share, which leads to higher competition and lower buyer surplus. In particular, this means they are more likely to use higher bid multipliers. In the second bullet point, by bidding aggressively above their true values, the bidder is also paying much more than they "should" in auctions where they are the rightful winner, which depletes their buyer surplus without hurting the efficiency (because they are the rightful winner in these auctions). We will prove below that this is in fact what happens.

In order to measure the balancedness of a market instance, we will aggregate the balancedness across slices via the balancedness quantile function defined as follows.

**Definition 3** (Balancedness quantile function). Fix any market instance $(n, m, s, \{S_k\}, \boldsymbol{v})$. The balancedness quantile function $F$ of the market instance is a function such that for any given $t \in [0, 1]$, $F(t)$ is the infimum value $b$ such that at least a $t$ fraction of the entire market has balancedness at most $b$. That is,

$$F(t) = \inf \left\{ b \;\middle|\; \sum_{k \in [s]: \mathsf{bal}_k \leq b} \mathsf{share}_k \geq t \right\}.$$

The balancedness quantile function can also be viewed as the inverse of the "cumulative distribution function" of the unbalancedness in the entire market. Note that $F(t) \in [0, 1/2]$ since $\mathsf{bal}_k \in [0, 1/2]$ whenever $n \geq 2$. Our actual bound will depend on the following parameter related to $F(t)$.

**Definition 4** (Unbalancedness of market instance). Fix any market instance $(n, m, s, \{S_k\}, \boldsymbol{v})$. Let $\alpha$ be the function such that for any $b \in [0, 1/2]$ and $u \in \mathbb{R}$,

$$\alpha(b, u) = 1 - 2b + \sqrt{b \cdot (1 - b)} \cdot u.$$

Moreover, let $\beta$ be the function such that for any $w \in [0, 1]$, $\beta(w)$ is the unique number $u$ such that,

$$\int_0^w \alpha(F(t), u) \, \mathrm{d}t = 1 - w.$$

The unbalancedness $\mathsf{unbal}(n, m, s, \{S_k\}, \boldsymbol{v})$ of the market instance is defined such that

$$\mathsf{unbal}(n, m, s, \{S_k\}, \boldsymbol{v}) =$$
$$\max_{w \in [0,1]} \int_0^w \left( 1 + \alpha(F(t), \beta(w)) - \sqrt{(1 - \alpha(F(t), \beta(w)))^2 + 4F(t) \cdot \alpha(F(t), \beta(w))} \right) \, \mathrm{d}t.$$

We establish the following intuitive properties of the unbalancedness, which will help make the conceptual messages of our main result (to be discussed in the next subsection) clearer.

**Proposition 1.** *The unbalancedness parameter has the following properties:*

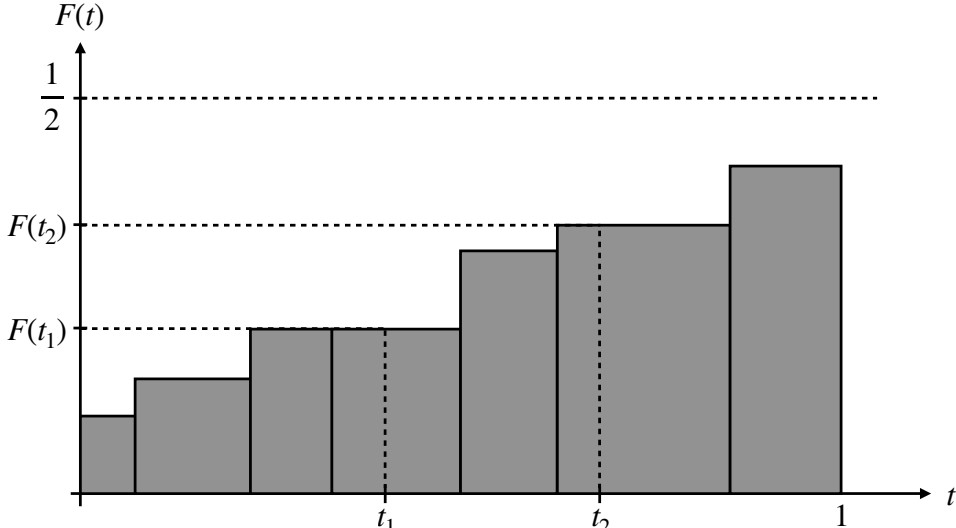

Figure 1: A graphical example of the balancedness quantile function. Each grey rectangle corresponds to a slice $k$, where the height is $\mathsf{bal}_k$ and the width $\mathsf{share}_k$. Observe that the height of each rectangle is at most $\mathsf{bal}_k \leq 1/2$, and the widths of all rectangles sum to $\sum_{k \in [s]} \mathsf{share}_k = 1$.

- $\mathsf{unbal}(n, m, s, \{S_k\}, \boldsymbol{v})$ *weakly increases when* $\mathsf{bal}_k$ *decreases for some slice* $k \in [s]$.
- $\mathsf{unbal}(n, m, s, \{S_k\}, \boldsymbol{v})$ *weakly increases when a slice is subdivided, i.e.,*

$$\mathsf{unbal}(n, m, s, \{S_k\}, \boldsymbol{v}) \leq \mathsf{unbal}(n, m, s+1, \{S_k'\}, \boldsymbol{v}),$$

  *where there exists* $k^* \in [s]$, *such that* $S_k' = S_k$ *for all* $k \in [s] \setminus \{k^*\}$ *and* $S_{s+1}' \cup S_{k^*}' = S_{k^*}$.[3]
- $\mathsf{unbal}(n, m, s, \{S_k\}, \boldsymbol{v}) \in [2/5, 1]$ *for each market instance* $(n, m, s, \{S_k\}, \boldsymbol{v})$ *where* $n \geq 2$ *and* $s \geq 2$.

*Proof.* Note that the unbalancedness depends only on the balancedness quantile function. We first argue that the unbalancedness weakly increases when the balancedness quantile function pointwise weakly decreases. Observe that $\mathsf{unbal}(\cdot)$ can be equivalently defined in the following way: For each $w \in [0, 1]$, let

$$\gamma(w) = \max_{\lambda \in \Lambda(w)} \int_0^w \left( 1 + \lambda(t) - \sqrt{(1 - \lambda(t))^2 + 4F(t) \cdot \lambda(t)} \right) \mathrm{d}t,$$

where $\Lambda(w)$ is the set of mappings from $[0, w]$ to $[0, \infty)$, satisfying for each $\lambda \in \Lambda(w)$,

$$\int_0^w \lambda(t) \, \mathrm{d}t = 1 - w.$$

Note that this maximum is guaranteed to exist.[4] Then

$$\mathsf{unbal}(n, m, s, \{S_k\}, \boldsymbol{v}) = \max_{w \in [0,1]} \gamma(w).$$

This alternative definition is equivalent to Definition 4 because for a fixed $w$, the maximizing $\lambda$ of $\gamma(\lambda)$ must satisfy: For each $t_1, t_2 \in [0, w]$

$$\frac{\partial(1 + \lambda(t_1) - \sqrt{(1 - \lambda(t_1))^2 + 4F(t_1)\lambda(t_1)})}{\partial(\lambda(t_1))} = \frac{\partial(1 + \lambda(t_2) - \sqrt{(1 - \lambda(t_2))^2 + 4F(t_2)\lambda(t_2)})}{\partial(\lambda(t_2))}.$$

---

[3]Here we do not assume slices are ordered such that the balancedness is weakly increasing.

[4]To quickly see why this is the case, observe that without loss of generality $\lambda$ is piecewise constant with at most $s + 1$ pieces, because $F$ is piecewise constant with at most $s$ pieces.

This must be true because otherwise one can adjust $\lambda$ locally and achieve a larger value of the integral. Given the above, one can show (see the proof of Lemma 1 in Appendix A for a detailed argument) that the maximizing $\lambda$ must satisfy: There exists some $u$ such that for each $t \in [0, w]$, $\lambda(t) = \alpha(F(t), u)$, where $\alpha$ is defined in Definition 4. The unique choice of $u$ that satisfies the constraint on $\lambda$ is then $\beta(w)$, and the alternative definition then reduces to Definition 4.

Now to argue that unbal weakly increases when $F$ pointwise weakly decreases, we only need to show that for each $w \in [0, 1]$, $\gamma(w)$ weakly increases when $F$ pointwise weakly decreases. The latter further reduces to: Fixing any $\lambda$ satisfying the conditions above, the integral

$$\int_0^w \left(1 + \lambda(t) - \sqrt{(1 - \lambda(t))^2 + 4F(t) \cdot \lambda(t)}\right) \mathrm{d}t$$

weakly increases when $F$ pointwise weakly decreases. This is true because the integrand increases when $F(t)$ decreases.

Now we come back to the properties to be proved. The first property becomes almost obvious, because when $\mathsf{bal}_k$ weakly decreases for some slice $k$, $F$ must pointwise weakly decrease, and therefore unbal must weakly increase. As for the third property, we consider extreme cases of $F$: When $F(t) = 0$ for each $t \in [0, 1]$, in Definition 4, $\alpha(F(t), u)$ is always 1, and $\mathsf{unbal}(n, m, s, \{S_k\}, \boldsymbol{v}) = 1$. This is the largest unbalancedness possible. When $F(t) = 1/2$ for each $t \in [0, 1]$, $\alpha(F(t), u) = u/2$, $\beta(w) = 2/w - 2$, and one can show $\mathsf{unbal}(n, m, s, \{S_k\}, \boldsymbol{v}) = 2/5$ (achieved when $w = 3/5$). This is the smallest unbalancedness possible.

The second property is a bit trickier. To see why this is true, consider the parameters $w$ and $\lambda$ that achieve $\mathsf{unbal}(n, m, s, \{S_k\}, \boldsymbol{v})$ in the alternative definition. Moreover, without loss of generality (recall that $\lambda$ is without loss of generality piecewise constant), suppose $\lambda$ is weakly decreasing on $\left(\sum_{k \in [k^*-1]} \mathsf{share}_k, \sum_{k \in [k^*]} \mathsf{share}_k\right)$. Let $\mathsf{share}'_k$ and $\mathsf{bal}'_k$ be the market share and balancedness of each slice $k$ in the new market instance. Without loss of generality, suppose $\mathsf{bal}'_{k^*} \le \mathsf{bal}'_{s+1}$. Define $G$ to be the "partially unordered" balancedness quantile function of the new market instance $(n, m, s + 1, \{S'_k\}, \boldsymbol{v})$ that preserves the ordering of the slices before subdivision, i.e.,

$$G(t) = \begin{cases} F(t), & \text{if } t \le \sum_{k \in [k^*-1]} \mathsf{share}_k \text{ or } t > \sum_{k \in [k^*]} \mathsf{share}_k \\ \mathsf{bal}'_{k^*}, & \text{if } \sum_{k \in [k^*-1]} \mathsf{share}_k < t \le \sum_{k \in [k^*]} \mathsf{share}'_k \\ \mathsf{bal}'_{s+1}, & \text{otherwise} \end{cases} .$$

Intuitively, $G$ is obtained by splitting slice $k^*$ "in place" from $F$. One can show that

$$\mathsf{unbal}(n, m, s + 1, \{S'_k\}, \boldsymbol{v}) \ge \int_0^w \left(1 + \lambda(t) - \sqrt{(1 - \lambda(t)^2 + 4G(t) \cdot \lambda(t)}\right) \mathrm{d}t.$$

This is essentially because the alternative definition is oblivious to "ordering" of the balancedness quantile function. So,

$$\mathsf{unbal}(n, m, s + 1, \{S'_k\}, \boldsymbol{v}) - \mathsf{unbal}(n, m, s, \{S_k\}, \boldsymbol{v})$$
$$\ge \int_{\sum_{k \in [k^*-1]} \mathsf{share}_k}^{\sum_{k \in [k^*]} \mathsf{share}_k} \left(\sqrt{(1 - \lambda(t))^2 + 4F(t) \cdot \lambda(t)} - \sqrt{(1 - \lambda(t))^2 + 4G(t) \cdot \lambda(t)}\right) \mathrm{d}t.$$

We only need to argue that the right hand side is at least 0. Since $\lambda$ is weakly decreasing, $F$ is constant, and $G$ is weakly increasing (because $\mathsf{bal}'_{k^*} \le \mathsf{bal}'_{s+1}$), in the worst case, $\lambda$ is constant on the interval of interest here (say $\lambda(t) = \ell > 0$). Then, the right hand side becomes

$$\mathsf{share}_{k^*} \cdot \sqrt{(1 - \ell)^2 + 4\mathsf{bal}_{k^*} \cdot \ell}$$
$$- \mathsf{share}'_{k^*} \cdot \sqrt{(1 - \ell)^2 + 4\mathsf{bal}'_{k^*} \cdot \ell} - \mathsf{share}'_{s+1} \cdot \sqrt{(1 - \ell)^2 + 4\mathsf{bal}'_{s+1} \cdot \ell}.$$

Here we have

$$\mathsf{share}'_{k^*} + \mathsf{share}'_{s+1} = \mathsf{share}_{k^*} \quad \text{and} \quad \mathsf{bal}'_{k^*} \cdot \mathsf{share}'_{k^*} + \mathsf{bal}'_{s+1} \cdot \mathsf{share}'_{s+1} \le \mathsf{bal}_{k^*} \cdot \mathsf{share}_{k^*},$$

because the two new slices are obtained by subdividing the old slice $k^*$. For brevity, let

$$a = \mathsf{share}'_{k^*}, \; b = \mathsf{share}'_{s+1}, \; c = \mathsf{share}_{k^*}, \; x = \mathsf{bal}'_{k^*}, \; y = \mathsf{bal}'_{s+1}, \; z = \mathsf{bal}_{k^*}.$$

We only need to show

$$c\sqrt{(1-\ell)^2 + 4z\ell} \geq a\sqrt{(1-\ell)^2 + 4x\ell} + b\sqrt{(1-\ell)^2 + 4y\ell},$$

where

$$a + b = c \quad \text{and} \quad ax + by \leq cz.$$

Applying Cauchy-Schwarz, we have

$$
\begin{aligned}
a\sqrt{(1-\ell)^2 + 4x\ell} + b\sqrt{(1-\ell)^2 + 4y\ell} &\leq \sqrt{(a+b)\cdot(a((1-\ell)^2 + 4xl) + b((1-\ell)^2 + 4y\ell))} \\
&= \sqrt{(a+b)^2(1-\ell)^2 + (a+b)(4ax + 4by)\ell} \\
&\leq \sqrt{c^2(1-\ell)^2 + 4c^2z\ell} \\
&= c\sqrt{(1-\ell)^2 + 4z\ell}.
\end{aligned}
$$

This finishes the proof. $\qquad\square$

## 3.2 Efficiency under the Balancedness Parametrization

Now we are ready to prove our main result, which links auction efficiency with unbalancedness when autobidders perform partially non-uniform bidding.

**Theorem 1** (PoA of FPA with partially non-uniform bidding). *For any $t \in [2/5, 1]$, we have*

$$\inf_{(n,m,s,\{S_k\},\boldsymbol{v}) \in \mathcal{I}_t} \mathsf{PoA}(n,m,s,\{S_k\},\boldsymbol{v}) = 1 - \frac{1}{2}t,$$

*where*

$$\mathcal{I}_t = \{(n,m,s,\{S_k\},\boldsymbol{v}) \mid \mathsf{unbal}(n,m,s,\{S_k\},\boldsymbol{v}) \leq t\}.$$

The theorem is a corollary of Lemma 1 and Lemma 2, which are stated and proved in Appendix A. Below we discuss the conceptual implications of Theorem 1.

**Better autobidders lead to worse efficiency.**  Theorem 1 and Proposition 1 together establish a possibly counterintuitive relation between the capability of autobidders and the efficiency of the auction outcome: Better optimized autobidders generally are capable of implementing more sophisticated non-uniform bidding strategies, which correspond to establishing finer partitions of the entire market into slices. However, by the second bullet of Proposition 1, any refinement of the slices can only lead to higher unbalancedness, which, by Theorem 1, leads to worse auction efficiency. In the extreme case where autobidders are capable of bidding optimally in each individual auction (which, in our model, corresponds to the case in which each auction forms its own slice with balancedness 0), the unbalancedness of the market is 1, and therefore, Theorem 1 states that the PoA of such a market is $1/2$, matching the bound established in prior work [Liaw et al., 2023, Deng et al., 2022]. In fact, this is precisely the phenomenon observed empirically by Deng et al. [2024]. Our results therefore provide a theoretical explanation for their empirical findings. Together with empirical results from [Deng et al., 2024], our theoretical results further suggest that it is unlikely to achieve better auction efficiency (especially in first-price marketplaces) by using autobidders that are (more) capable of non-uniform bidding in practice.

**The importance of balanced channels / slices.**  In situations where there are naturally multiple channels or slices (e.g., different ad exchanges operated by different companies, or by different organizations within a single company) [Deng et al., 2023a, Susan et al., 2023], our results also highlight the importance of keeping each channel / slice balanced. In particular, the first bullet of Proposition 1 suggests that if the balancedness of one slice can be improved without hurting the balancedness of the other slices, then the unbalancedness of the entire market decreases, and Theorem 1 guarantees better efficiency overall. In practice, our results suggest that it is better off for different channels / slices to coordinate to achieve higher overall efficiency.

**Even perfectly balanced slices introduce inefficiency.**  Proposition 1 states that whenever there are at least 2 slices in the market instance, the unbalancedness of the market instance is at least $2/5$, even if all slices are perfectly balanced. Theorem 1 then implies that the efficiency of such market instances is at most $4/5$, which suggests that efficiency loss is inevitable whenever the market consists

of multiple slices. In contrast, when autobidders perform uniform bidding over the entire market (i.e., when there is only $1$ slice), it is known that first-price marketplaces achieve full efficiency [Balseiro et al., 2021a]. In other words, this indicates that one may expect to see a phase transition in market efficiency when moving away from uniform bidding to (even highly restrictive) non-uniform bidding. We also remark that for second-price marketplaces, the PoA is $2$ even with perfectly balanced slices, or a single slice. This suggests that first-price marketplaces generally achieve higher efficiency compared to second-price ones.

## Acknowledgments and Disclosure of Funding

We thank anonymous reviewers for their helpful feedback.

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

# A Proof of Theorem 1

**Lemma 1.** *For each market instance* $(n, m, s, \{S_k\}, \boldsymbol{v})$,

$$\mathsf{PoA}(n, m, s, \{S_k\}, \boldsymbol{v}) \geq 1 - \frac{1}{2}\mathsf{unbal}(n, m, s, \{S_k\}, \boldsymbol{v}).$$

*Proof.* Fix a market instance $(n, m, s, \{S_k\}, \boldsymbol{v})$, together with an equilibrium given by bid multipliers $\boldsymbol{\theta}$. Without loss of generality, suppose the optimal liquid welfare is 1, i.e.,

$$\sum_{j \in [m]} \max_{i \in [n]} v_{i,j} = 1.$$

For each bidder $i \in [n]$ and auction $j \in [m]$, let the "loss of efficiency" $\mathsf{loss}_{i,j}$ caused by $i$ in $j$ be:

$$\mathsf{loss}_{i,j} = x_{i,j} \cdot (v_{\mathsf{rw}(j),j} - v_{i,j}).$$

Moreover, abusing notation, for each bidder $i \in [n]$ and each slice $k \in [s]$, let

$$\mathsf{loss}_{i,k} = \sum_{j \in S_k} \mathsf{loss}_{i,j} \quad \text{and} \quad \mathsf{loss}_k = \sum_{i \in [n]} \mathsf{loss}_{i,k}.$$

We only need to show

$$\sum_{i \in [n], j \in [m]} v_{i,j} \cdot x_{i,j} \geq 1 - \frac{1}{2}\mathsf{unbal}(n, m, s, \{S_k\}, \boldsymbol{v}),$$

which is equivalent to

$$\sum_{k \in [s]} \mathsf{loss}_k \leq \frac{1}{2}\mathsf{unbal}(n, m, s, \{S_k\}, v).$$

To upper bound the total loss of efficiency, we first establish a tradeoff between the total loss on slice $k$ and the total "overpayment" on the same slice. This will involve upper bounding the loss and lower bounding the overpayment. Fix a slice $k$ and partition auctions on $k$ into 3 sets:

- $A_k$: auctions where the rightful winner wins with a bid multiplier smaller than 1, i.e.,
$$A_k = \{j \in S_k \mid x_{\mathsf{rw}(j),j} = 1 \text{ and } \theta_{\mathsf{rw}(j),k} < 1\};$$

- $B_k$: auctions where the rightful winner wins with a bid multiplier larger than or equal to 1, i.e.,
$$B_k = \{j \in S_k \mid x_{\mathsf{rw}(j),j} = 1 \text{ and } \theta_{\mathsf{rw}(j),k} \geq 1\};$$

- $C_k$: auctions where the rightful winner does not win, i.e.,
$$C_k = \{j \in S_k \mid x_{\mathsf{rw}(j),j} = 0\}.$$

For brevity, we define the "reserved" share $\mathsf{res}_k$ to be the market share of set $A_k$, i.e., $\mathsf{res}_k = \mathsf{share}_{A_k} \leq \mathsf{share}_k$.

Consider any bidder $i$. First observe that for each auction $j \in S_k$, $\mathsf{loss}_{i,j} > 0$ only if the following happen simultaneously:

- Bidder $i$ must outbid the rightful winner in auction $j$, i.e., $p_{i,j} = b_{i,j} \geq b_{\mathsf{rw}(j),j}$.

- Since the rightful winner $\mathsf{rw}(j)$ is not winning in $j$, we must have $\theta_{\mathsf{rw}(j),k} \geq 1$, which implies $b_{\mathsf{rw}(j),j} \geq v_{\mathsf{rw}(j),j}$.

Given the above properties, for each auction $j \in S_k$ where $\mathsf{loss}_{i,j} > 0$, we have $x_{i,j} = 1$ and $\theta_{i,k} \cdot v_{i,j} = b_{i,j} \geq b_{\mathsf{rw}(j),j} \geq v_{\mathsf{rw}(j),j}$. As a result,

$$\mathsf{loss}_{i,j} = v_{\mathsf{rw}(j),j} - v_{i,j} \leq v_{\mathsf{rw}(j),j} - v_{\mathsf{rw}(j),j}/\theta_{i,k} = \frac{\theta_{i,k} - 1}{\theta_{i,k}} \cdot v_{\mathsf{rw}(j),j} \leq \frac{\theta_{i^*,k} - 1}{\theta_{i^*,k}} \cdot v_{rw(j),j},$$

where $i^* \in [n]$ is the bidder $i^*$ who has the largest bid multiplier on slice $k$. Recall that $C_k$ is the set of auctions in $S_k$ where $\sum_i \mathsf{loss}_{i,j} > 0$. Since for any auction $j \in S_k$ where $\mathsf{rw}(j) = i^*$, $x_{i^*,j} = 1$, we must have

$$\mathsf{share}_{C_k} \leq \mathsf{share}_k - \mathsf{res}_k - \mathsf{share}_{i^*,k} \leq (1 - \mathsf{bal}_k) \cdot \mathsf{share}_k - \mathsf{res}_k.$$

Summing the upper bound for $\mathsf{loss}_{i,j}$ over $i \in [n]$ and $j \in S_k$, and plugging in the above inequality involving $\mathsf{share}_{C_k}$, we get

$$
\begin{aligned}
\mathsf{loss}_k &= \sum_{i \in [n], j \in S_k : \mathsf{loss}_{i,j} > 0} \mathsf{loss}_{i,j} \\
&\leq \sum_{j \in C_k} \frac{\theta_{i^*,k} - 1}{\theta_{i^*,k}} \cdot v_{\mathsf{rw}(j),j} \\
&= \frac{\theta_{i^*,k} - 1}{\theta_{i^*,k}} \cdot \mathsf{share}_{C_k} \\
&\leq \frac{\theta_{i^*,k} - 1}{\theta_{i^*,k}} \cdot ((1 - \mathsf{bal}_k) \cdot \mathsf{share}_k - \mathsf{res}_k).
\end{aligned}
$$

This is our upper bound on the total loss, which will be useful later.

Now we lower bound the total overpayment. Note that $x_{i,j} = 1$ implies $b_{i,j} = p_{i,j}$. So whenever $\mathsf{loss}_{i,j} > 0$, based on the observations in the two bullet points above, we must have

$$\mathsf{loss}_{i,j} \leq x_{i,j} \cdot (b_{\mathsf{rw}(j),j} - v_{i,j}) \leq x_{i,j} \cdot (b_{i,j} - v_{i,j}) = x_{i,j} \cdot (p_{i,j} - v_{i,j}) = p_{i,j} - x_{i,j} \cdot v_{i,j}.$$

In words, $i$ can only cause as much loss of efficiency in $j$ as the amount $i$ overpays in $j$. To this end, let

$$\mathsf{sur}_{i,j} = x_{i,j} \cdot v_{i,j} - p_{i,j}$$

be the surplus $i$ gets from $j$. We have: for each $i \in [n]$ and $j \in [m]$,

$$\mathsf{loss}_{i,j} \leq \max\{0, -\mathsf{sur}_{i,j}\}.$$

Summing over $i \in [n]$ and $j \in C_k$, we get

$$\sum_{i \in [n], j \in L_k} \max\{0, -\mathsf{sur}_{i,j}\} \geq \sum_{i \in [n], j \in L_k} \mathsf{loss}_{i,j} = \mathsf{loss}_k.$$

This is one of the two inequalities that we will use to lower bound the total overpayment.

On the other hand, consider the bidder $i^*$ who has the largest bid multiplier on slice $k$. Recall that for any auction $j \in S_k$ where $\mathsf{rw}(j) = i^*$, $x_{i^*,j} = 1$. This means $i^*$ overpays in auctions where $i^*$ is the rightful winner without causing any loss of efficiency. In fact, the amount of overpayment here can be bounded as

$$
\begin{aligned}
\sum_{j \in S_k : \mathsf{rw}(j) = i^*} -\mathsf{sur}_{i^*,j} &= \sum_{j \in S_k : \mathsf{rw}(j) = i^*} (\theta_{i^*,k} - 1) \cdot v_{i^*,j} \\
&= (\theta_{i^*,k} - 1) \cdot \mathsf{share}_{i^*,k} \\
&\geq (\theta_{i^*,k} - 1) \cdot \mathsf{share}_k \cdot \mathsf{bal}_k.
\end{aligned}
$$

So, combining this inequality with the one above, we get

$$\sum_{i \in [n], j \in S_k} \max\{0, -\mathsf{sur}_{i,j}\} \geq (\theta_{i^*,k} - 1) \cdot \mathsf{share}_k \cdot \mathsf{bal}_k + \mathsf{loss}_k.$$

This is the desired lower bound on the total overpayment.

Now let us put the bounds for the loss and that for the overpayment together into a tradeoff between the two quantities. For brevity, let

$$\mathsf{sur}_k^- = \sum_{i \in [n], j \in S_k} \max\{0, -\mathsf{sur}_{i,j}\}.$$

Recall the upper bound we have on the total loss:

$$\mathsf{loss}_k \leq \frac{\theta_{i^*,k} - 1}{\theta_{i^*,k}} \cdot ((1 - \mathsf{bal}_k) \cdot \mathsf{share}_k - \mathsf{res}_k).$$

This gives us

$$\theta_{i^*,k} \geq \frac{(1 - \mathsf{bal}_k) \cdot \mathsf{share}_k - \mathsf{res}_k}{(1 - \mathsf{bal}_k) \cdot \mathsf{share}_k - \mathsf{res}_k - \mathsf{loss}_k}.$$

Plugging this into the lower bound we have on the total overpayment, we get

$$\mathsf{sur}_k^- \geq \frac{\mathsf{loss}_k}{(1 - \mathsf{bal}_k) \cdot \mathsf{share}_k - \mathsf{res}_k - \mathsf{loss}_k} \cdot \mathsf{share}_k \cdot \mathsf{bal}_k + \mathsf{loss}_k$$

$$\implies \mathsf{loss}_k^2 - (\mathsf{share}_k - \mathsf{res}_k + \mathsf{sur}_k^-) \cdot \mathsf{loss}_k + ((1 - \mathsf{bal}_k) \cdot \mathsf{share}_k - \mathsf{res}_k) \cdot \mathsf{sur}_k^- \geq 0.$$

The latter is a quadratic inequality in $\mathsf{loss}_k$, which in our case implies

$$\mathsf{loss}_k \leq \frac{1}{2} \left( \mathsf{share}_k - \mathsf{res}_k + \mathsf{sur}_k^- - \sqrt{(\mathsf{share}_k - \mathsf{res}_k - \mathsf{sur}_k^-)^2 + 4\mathsf{share}_k \cdot \mathsf{bal}_k \cdot \mathsf{sur}_k^-} \right).$$

The above tradeoff is useful because together with an additional constraint on $\{\mathsf{res}_k\}$ and $\{\mathsf{sur}_k^-\}$, it gives an upper bound on $\sum_k \mathsf{loss}_k$, which is the quantity we want to bound. The additional constraint is from bidders' RoS constraints. For each bidder $i$, the RoS constraint requires that $\sum_{j \in [m]} \mathsf{sur}_{i,j} \geq 0$, which implies

$$\sum_{i \in [n], j \in [m]} \mathsf{sur}_{i,j} \geq 0.$$

On the other hand, for each auction $j$, $\mathsf{sur}_{i,j} > 0$ only if $j \in A_{\mathsf{slice}(j)}$ (i.e., $\theta_{\mathsf{rw}(j),\mathsf{slice}(j)} < 1$). This means

$$\sum_{i \in [n], j \in [m]} \max\{0, \mathsf{sur}_{i,j}\} \leq \sum_{k \in [s]} \mathsf{share}_{A_k} = \sum_{k \in [s]} \mathsf{res}_k.$$

As a result,

$$\sum_{k \in [s]} \mathsf{sur}_k^- \leq \sum_{i \in [n], j \in [m]} \max\{0, \mathsf{sur}_{i,j}\} \leq \sum_{k \in [s]} \mathsf{res}_k.$$

This is the additional constraint we need.

Now the problem of upper bounding $\sum_{k \in [s]} \mathsf{loss}_k$ reduces to solving the following optimization problem (replacing all variables with shorthands: $x_k$ for $\mathsf{res}_k$, $y_k$ for $\mathsf{sur}_k^-$, and $z_k$ for $\mathsf{loss}_k$):

$$\max \quad \sum_{k \in [s]} z_k$$

$$\text{s.t.} \quad \forall k \in [s]: \ z_k \leq \frac{1}{2} \left( \mathsf{share}_k - x_k + y_k - \sqrt{(\mathsf{share}_k - x_k - y_k)^2 + 4\mathsf{share}_k \cdot \mathsf{bal}_k \cdot y_k} \right)$$

$$\sum_{k \in [s]} y_k \leq \sum_{k \in s} x_k$$

$$\forall k \in [s]: \ 0 \leq x_k \leq \mathsf{share}_k$$

$$\forall k \in [s]: \ y_k \geq 0.$$

The rest of the proof is devoted to solving this optimization problem.

First we make some structural observations. Let $f_k(x_k, y_k)$ be the right hand side of the upper bound on $z_k$. When optimality is achieved:

- $z_k = f_k(x_k, y_k)$ for all $k \in [s]$, and $\sum_{k \in [s]} y_k = \sum_{k \in [s]} s_k$. The latter is simply because $f_k(x_k, y_k)$ is strictly increasing in $y_k$.

- For each $k \in [s]$, if $x_k \geq (1 - \mathsf{bal}_k) \cdot \mathsf{share}_k$, then $y_k = 0$ and $x_k = \mathsf{share}_k$. This is because in such cases, $f_k(x_k, y_k) \leq 0$, and $f_k(x_k, y_k) = 0$ when $y_k = 0$. So we would save our budget for $\{y_{k'}\}$ on slice $k$ and spend it elsewhere. For similar reasons, we would max out $x_k$ to create more budget for $\{y_{k'}\}$. In particular, this property implies that $x_k = \mathsf{share}_k$ and $y_k > 0$ cannot happen simultaneously.

With the above observations enforced as constraints (which does not change the optimal objective value), we further relax the problem. In particular, for each slice $k$, we replace the constraint that $z_k \leq f_k(x_k, y_k)$ with $z_k \leq g_k(x_k, y_k)$, where

$$g_k(x_k, y_k) = \frac{1}{2}\left(\mathsf{share}_k - x_k + y_k - \sqrt{(\mathsf{share}_k - x_k - y_k)^2 + 4(\mathsf{share}_k - x_k) \cdot \mathsf{bal}_k \cdot y_k}\right).$$

This relaxation essentially means we imagine slices are arbitrarily divisible, and we can divide each slice into two with the same balancedness, whose market shares sum to that of the one before division. As we will see, the relaxation has no cost under our parametrization, because the hard market instances are precisely the ones that divide slices in the way that achieves the maximum objective value in the relaxed problem. We consider this relaxed problem with the additional constraints (in the bullet points) from now on.

For this relaxed problem, we make more structural observations when optimality is achieved:

- There is at most one slice $k$ where $0 < x_k < \mathsf{share}_k$. This is because fixing $\sum_{k \in [s]} x_k$, the most efficient way to distribute this sum into slices is greedy. That is, we first find the last slice $s$ which has the minimum balancedness $\mathsf{bal}_s$ and increase $x_s$. If we reach the limit, i.e., $x_s = \mathsf{share}_s$, then we move on to the next slice $s - 1$ among the remaining ones with the minimum balancedness and repeat this, etc., until we reach the desired sum. One may check that this procedure in fact induces the sub-optimization problem over $\{y_k\}$ and $\{z_k\}$ with the largest optimal objective value given $\sum_{k \in [s]} x_k$.

- For any $k, k' \in [s]$ where $\min\{y_k, y_{k'}\} > 0$, we must have

$$\frac{\partial g_k(x_k, y_k)}{\partial y_k} = \frac{\partial g_{k'}(x_{k'}, y_{k'})}{\partial y_{k'}},$$

because otherwise we can locally adjust $y_k$ and $y_{k'}$ (while keeping the sum unchanged) and get a strictly larger objective value.

Putting the above observations together, we conclude that the optimal solution to the optimization problem must have the following structure: There is an integer $s' \in [s]$, a positive real number $C \in [0, 1]$, and "effective" market shares $\{\mathsf{share}'_k\}$, such that:

- $\mathsf{share}'_k = \mathsf{share}_k$ for each $k \in [s' - 1]$, $\mathsf{share}'_{s'} \leq \mathsf{share}_{s'}$, and

$$\sum_{k \in [s']} \mathsf{share}'_k = C.$$

- $x_k = \mathsf{share}_k - \mathsf{share}'_k$ for each $k \in [s]$.

- There exists a real number $D > 0$ such that $\frac{\mathrm{d}h_k(y_k)}{\mathrm{d}y_k} = D$ for each $k \in [s']$, where

$$h_k(y_k) = \frac{1}{2}\left(\mathsf{share}'_k + y_k - \sqrt{(\mathsf{share}'_k - y_k)^2 + 4\mathsf{share}'_k \cdot \mathsf{bal}_k \cdot y_k}\right).$$

In particular, fixing $C$, $y_k/\mathsf{share}'_k$ should only depend on $\mathsf{bal}_k$.

- For each $k > s'$, $y_k = 0$, and

$$\sum_{k \in [s']} y_k = 1 - C.$$

Below we solve the relaxed problem given all simplifying observations above. The key step is to pin down the dependency of $y_k/\text{share}'_k$ on $\text{bal}_k$ given $C$. In fact, for each $k \in [s']$,

$$\exists D, \forall k \in [s'], \frac{\mathrm{d}h_k(y_k)}{\mathrm{d}y_k} = D$$

$$\implies \exists D, \forall k \in [s'], \frac{(y_k - \text{share}'_k) + 2\text{share}'_k \cdot \text{bal}_k}{\sqrt{(y_k - \text{share}'_k)^2 + 4\text{share}'_k \cdot \text{bal}_k \cdot y_k}} = D$$

$$\implies \exists D, \forall k \in [s'], \frac{(y_k - \text{share}'_k)^2 + 4\text{share}'_k \cdot \text{bal}_k \cdot (y_k - \text{share}'_k) + 4\text{share}'^2_k \cdot \text{bal}^2_k}{(y_k - \text{share}'_k)^2 + 4\text{share}'_k \cdot \text{bal}_k \cdot y_k} = D$$

$$\implies \exists D, \forall k \in [s'], \frac{\text{share}'^2_k \cdot \text{bal}_k \cdot (1 - \text{bal}_k)}{(y_k - \text{share}'_k)^2 + 4\text{share}'_k \cdot \text{bal}_k \cdot y_k} = D$$

$$\implies \exists D, \forall k \in [s'], \frac{(y_k - \text{share}'_k)^2 + 4\text{share}'_k \cdot \text{bal}_k \cdot y_k}{\text{share}'^2_k \cdot \text{bal}_k \cdot (1 - \text{bal}_k)} = D$$

$$\implies \exists D, \forall k \in [s'], y_k = \text{share}'_k \cdot \left( (1 - 2\text{bal}_k) \pm \sqrt{\text{bal}_k \cdot (1 - \text{bal}_k) \cdot (D - 4)} \right)$$

$$\implies \exists D, \forall k \in [s'], y_k = \text{share}'_k \cdot \left( (1 - 2\text{bal}_k) \pm \sqrt{\text{bal}_k \cdot (1 - \text{bal}_k) \cdot D} \right).$$

Moreover, observe that the sign in the above expression must be the same for all $k \in [s']$ in order for the derivatives to be the same, which means

$$\exists D \in \mathbb{R}, \forall k \in [s'], y_k = \text{share}'_k \cdot \left( (1 - 2\text{bal}_k) + \sqrt{\text{bal}_k \cdot (1 - \text{bal}_k) \cdot D} \right).$$

Abusing notation, let

$$y_k(D) = \text{share}'_k \cdot \left( (1 - 2\text{bal}_k) + \sqrt{\text{bal}_k \cdot (1 - \text{bal}_k) \cdot D} \right).$$

Note that here $D$ can be either positive or negative.

Given the above observation, when optimality is achieved, the solution is essentially parametrized by the parameter $C$. In particular, $s'$ and $\{\text{share}'\}$ are uniquely determined by $C$, and the parameter $D$ is unique given $C$.[5] In fact, since $y_k(D)$ is monotone for each $k \in [s']$, there is a unique $D$ such that

$$\sum_{k \in [s']} y_k(D) = 1 - C.$$

So we may alternatively write $y_k(C) = y_k(D(C))$ for the unique choice of $y_k$ given $C$. Below we link the maximum value of the objective to the unbalancedness of the market instance and conclude the proof.

Recall that

$$\alpha(b, u) = 1 - 2b + \sqrt{b \cdot (1 - b)} \cdot u.$$

So

$$y_k(C) = \text{share}'_k \cdot \alpha(\text{bal}_k, D(C)).$$

Note that here $\text{share}'_k$ depends on $C$, and we omit this dependency for brevity. Given the choice of $D(C)$, since the slices are indexed such that $\text{bal}_k$ is weakly increasing, we must have

$$\int_0^C \alpha(F(t), D(C)) \, \mathrm{d}t = \sum_{k \in [s']} \alpha(\text{bal}_k, D(C)) \cdot \text{share}'_k = \sum_{k \in [s']} y_k(C) = 1 - C,$$

where $F(t)$ is the balancedness quantile function of the market instance. In other words, the choice of $D$ given $C$ is precisely $D(C) = \beta(C)$ as defined in Definition 4. Then the objective can be bounded

---

[5]Strictly speaking, this is not true when $\text{bal}_k = 0$ for all $k \in [s']$. However, in such case, the choice of $D$ does not matter, and one must choose $C$ to satisfy the constraint on $\sum_{k \in [s']} y_k$. In the rest of the proof, we omit further discussion as this is a relatively straightforward corner case, and simply assume $D$ is unique given $C$.

as

$$\sum_{k \in [s]} h_k(y_k)$$

$$= \sum_{k \in [s]} \frac{1}{2} \left( \mathsf{share}'_k + y_k - \sqrt{(\mathsf{share}'_k - y_k)^2 + 4\mathsf{share}'_k \cdot \mathsf{bal}_k \cdot y_k} \right)$$

$$= \sum_{k \in [s']} \frac{1}{2}\mathsf{share}'_k \cdot \left( 1 + y_k/\mathsf{share}'_k - \sqrt{(1 - y_k/\mathsf{share}'_k)^2 + 4\mathsf{bal}_k \cdot y_k/\mathsf{share}'_k} \right)$$

$$= \sum_{k \in [s']} \frac{1}{2}\mathsf{share}'_k \cdot \left( 1 + \alpha(\mathsf{bal}_k, \beta(C)) - \sqrt{(1 - \alpha(\mathsf{bal}_k, \beta(C)))^2 + 4\mathsf{bal}_k \cdot \alpha(\mathsf{bal}_k, \beta(C))} \right)$$

$$= \frac{1}{2} \int_0^C \left( 1 + \alpha(F(t), \beta(C)) - \sqrt{(1 - \alpha(F(t), \beta(C)))^2 + 4F(t) \cdot \alpha(F(t), \beta(C))} \right) \, \mathrm{d}t$$

$$\leq \frac{1}{2} \max_{w \in [0,1]} \int_0^w \left( 1 + \alpha(F(t), \beta(w)) - \sqrt{(1 - \alpha(F(t), \beta(w)))^2 + 4F(t) \cdot \alpha(F(t), \beta(w))} \right) \, \mathrm{d}t$$

$$= \frac{1}{2}\mathsf{unbal}(n, m, s, \{S_k\}, \boldsymbol{v}).$$

This concludes the proof of the lemma. $\square$

**Lemma 2.** *For each $t \in [2/5, 1]$, there exists a market instance $(n, m, s, \{S_k\}, \boldsymbol{v})$ which satisfies* $\mathsf{unbal}(n, m, s, \{S_k\}, \boldsymbol{v}) = t$ *and* $\mathsf{PoA}(n, m, s, \{S_k\}, \boldsymbol{v}) = 1 - t/2$.

*Proof.* Recall that the proof of Proposition 1 establishes that $\mathsf{unbal}(F)$ (abusing notation here since unbal depends only on $F$) weakly increases whenever $F$ pointwise weakly decreases. Moreover, $\mathsf{unbal}(F)$ is clearly continuous in $F$ (where we use any natural norm for the space where $F$ resides, e.g., $L_1$). These properties together ensures that for any $t \in [2/5, 1]$, there exists some $F$ such that $\mathsf{unbal}(F) = t$ — in fact, there exists a constant function $F$ such that $\mathsf{unbal}(F) = t$, i.e., there exists $f \in [0, 1/2]$ such that the function $F$ where $F(x) = f$ for all $x \in [0, 1]$ satisfies $\mathsf{unbal}(F) = t$. So, we only need to show that given any constant function $F$ where (1) $F(x)$ is constantly $f$ for some $f \in [0, 1/2]$ and (2) $\mathsf{unbal}(F) = t$, there exists a market instance $(n, m, s, \{S_k\}, \boldsymbol{v})$ whose balancedness quantile function is $F$, and $\mathsf{PoA}(n, m, s, \{S_k\}, \boldsymbol{v}) \leq 1 - t/2$.

We construct such a market instance with $n = 3$ bidders, $m = 6$ auctions, and $s = 3$ slices where each slice contains precisely 2 auctions. Let $S_1 = \{1, 2\}$, $S_2 = \{3, 4\}$, and $S_3 = \{5, 6\}$. Intuitively, we will construct valuations together with an equilibrium, where (1) on $S_1$, bidder 1 steals from bidder 3, (2) on $S_2$, bidder 2 steals from bidder 3, and (3) on $S_3$, both bidder 1 and bidder 2 win their market share without any competition. Below we choose the exact valuations that implement this plan.

Let $w \in [0, 1]$ be a parameter to be optimized later. The valuations we choose depend on $f$ and $w$. Below we will only specify the non-zero part of the valuations. Consider $S_3$ first: We let $v_{1,5} = f \cdot (1 - w)$ and $v_{2,6} = (1 - f) \cdot (1 - w)$. On slice $S_1$, we let $v_{1,1} = f^2 \cdot w$, $v_{3,2} = f \cdot (1 - f) \cdot w$, and $v_{1,2}$ be such that

$$\left( \frac{v_{3,2}}{v_{1,2}} - 1 \right) \cdot (v_{1,1} + v_{1,2}) = f \cdot (1 - w).$$

In particular, the choice of $v_{1,2}$ guarantees that when bidder 1 pays precisely the minimum amount to "steal" the entire market share of bidder 3 on $S_1$, bidder 1's overall buyer surplus is 0. Similarly, on slice $S_2$, we let $v_{2,3} = f \cdot (1 - f) \cdot w$, $v_{3,4} = (1 - f)^2 \cdot w$, and $v_{2,4}$ be such that

$$\left( \frac{v_{3,4}}{v_{2,4}} - 1 \right) \cdot (v_{2,3} + v_{2,4}) = (1 - f) \cdot (1 - w).$$

In particular, the choice of $v_{2,4}$ guarantees that when bidder 2 pays precisely the minimum amount to "steal" the entire market share of bidder 3 on $S_2$, bidder 2's overall buyer surplus is 0.

Now observe that the following bidding strategies form an equilibrium: $\theta_{1,1} = v_{3,2}/v_{1,2}$, $\theta_{2,2} = v_{3,4}/v_{2,4}$, and all other bid multipliers are $1$.[6] This means

$$\mathsf{PoA}(n, m, s, \{S_k\}, \boldsymbol{v}) \leq \frac{v_{1,1} + v_{1,2} + v_{2,3} + v_{2,4} + v_{1,5} + v_{2,6}}{\mathsf{share}_1 + \mathsf{share}_2 + \mathsf{share}_3}$$
$$= \mathsf{share}_1 - h_1(f \cdot (1-w)) + \mathsf{share}_2 - h_2((1-f) \cdot (1-w)) + \mathsf{share}_3.$$

Recall that we still have the freedom to choose $w \in [0, 1]$. So we only need to show

$$\max_w (h_1(f \cdot (1-w)) + h_2((1-f) \cdot (1-w))) = \frac{1}{2}\mathsf{unbal}(n, m, s, \{S_k\}, \boldsymbol{v}).$$

In fact, we show that for each $w \in [0, 1]$,

$$h_1(f \cdot (1-w)) + h_2((1-f) \cdot (1-w)) = \max_{\lambda \in \Lambda(w)} \int_0^w \left(1 + \lambda(\tau) - \sqrt{(1-\lambda(\tau))^2 + 4f \cdot \lambda(\tau)}\right) d\tau.$$

Here, $h_1$ and $h_2$ are as defined in the proof of Lemma 1. This implies the above equation given the alternative definition of unbal established in the proof of Proposition 1. To see why this equation holds, observe that the right hand side is equal to

$$\max_{x \in [0,1]} h_1(x \cdot (1-w)) + h_2((1-x) \cdot (1-w)).$$

Moreover, since $\mathsf{bal}_1 = \mathsf{bal}_2 = f$, the optimizer must satisfy $x/\mathsf{share}_1 = (1-x)/\mathsf{share}_2$, which means $x = f$. This concludes the proof of the lemma. $\qquad\square$

---

[6]Here we assume a particular tiebreaking rule. The proof works without the assumption since one can perturb the valuations to avoid tiebreaking.

