# OpenReview forum: "Autobidder's Dilemma: Why More Sophisticated Autobidders Lead to Worse Auction Efficiency"
_NeurIPS.cc/2024/Conference — NeurIPS 2024 poster_

### Official Review · Reviewer_uz4Y · 2024-07-09

**Soundness:** 4
**Presentation:** 3
**Contribution:** 3
**Rating:** 5
**Confidence:** 4

**Summary:**

The paper provide a fine-grained price of anarchy analysis for autobidders with non-uniform bid-scaling strategies in first price auctions, showing that first price auctions are more efficient when autobidders are less powerful and are more efficient with more balanced slices.

**Strengths:**

1. Concrete theoretical analysis for PoA of first price auctions with non-uniform autobidders.
2. It is interesting to find the definition of balancedness of auctions, which can be regarded as a kind of power of auctioneers, are highly related to the efficiency of auctions w.r.t. autobidders.

**Weaknesses:**

1. The main conclusions of the paper are indeed unsurprising, although it is good to know with a theoretical proof. This does weaken the contribution of this paper.
2. The framework of flice-based model is kind of restrictive. It is hard to imagine that bidders in real-world auctions bid uniformly in each slides partitioned by the auctioneer.

**Questions:**

NA

---

> ### Author Rebuttal · Authors · 2024-08-06
>
> Thank you for your insightful and detailed comments.
>
> **Slice-based model**: first we'd like to note that our model is consistent with prior work on non-uniform bid scaling, e.g., "Non-uniform Bid-scaling and Equilibria for Different Auctions: An Empirical Study" by researchers at Google.  In light of this, we believe this model already captures the essence of non-uniform bidding in reality.  For example, our messages might inform a platform's decision of merging two channels or (not) splitting a channel into several, whenever practically feasible.  On the other hand, another important consideration here is the tradeoff between the practicality of the model and the cleanliness of the messages.  In principle, one could certainly consider more general / sophisticated models and try to derive similar results there (e.g., we sketch one such possibility in our response to Reviewer TSHw), but doing so would probably make the messages much more obscure.  To this end, we'd like to argue that our approach achieves a reasonable balance on the spectrum.
>
> **Conclusions / messages of this paper**: We position our paper as a paper providing a theoretical explanation of the empirical findings from "Non-uniform Bid-scaling and Equilibria for Different Auctions: An Empirical Study". The conclusion might seem unsurprising provided the previously known empirical results but we’d like to argue that it is indeed technically non-trivial to develop such a theoretical explanation. Our result is also more interpretable and reliable (in the sense that there is a proof) than empirical ones regarding the same phenomenon.

---

> > ### Comment · Reviewer_uz4Y · 2024-08-13
> >
> > Thanks for your response.

---

### Official Review · Reviewer_DtQM · 2024-07-13

**Soundness:** 2
**Presentation:** 1
**Contribution:** 3
**Rating:** 4
**Confidence:** 2

**Summary:**

The paper considers the efficiency of auto bidders in first price auctions that use different shading factors on different slices of the items. Claims to show that the improved efficiency of the multi-slice system yields worse social welfare in equilibrium.

**Strengths:**

The topic of auto-bidding and efficiency of the results is extremely interesting.

**Weaknesses:**

The paper fails to clearly identify the model studied.

based on the rebuttal and the other reviewers comments, i now understand the model. Like reviewer TSHw I am not convinced at the reasonableness of this special case but will increase my score a bit.

**Questions:**

Please help me identify the model considered. what is the assumption about other bidders? Do they also use slices? and are they using the same slices? and what is the model of  the value of arriving items? Independent Bayesian? worst case?

**Limitations:**

No negative impact

---

> ### Author Rebuttal · Authors · 2024-08-06
>
> Thank you for your comments.
>
> In words: all bidders are symmetric and use the same slices, which are given exogenously.  Other than slices, we use the standard multi-item model, where essentially all items arrive at once at the very beginning, with the values given exogenously and publicly known (whether the values are public does not affect the behavior of the bidders, since they are responding to the *bids* of other bidders).  Each bidder picks a bid multiplier for each slice, so each bidder's strategy is a vector of bid multipliers.  We then consider equilibrium strategy profiles in this perfect-information environment, where each bidder's strategy must maximize their own total value subject to the ROI constraint, given all other bidders' strategies.  Our measure of efficiency is the price of anarchy, i.e., the welfare under the worst equilibrium, divided by the first-best optimal welfare.  We hope the above informal explanation can help clarify the reviewer's questions.

---

> > ### Comment · Reviewer_DtQM · 2024-08-11
> > **thank you for your clarification**
> >
> > Thank you for your clarification. Unfortunately, I agree with reviewer TSHw that the suggestion that all bidders use the same slices is very unnatural (even if other previous papers used it). Please at least clearly state this and explain why your work is limited to this case.

---

> > > ### Author Response · Authors · 2024-08-13
> > >
> > > Thank you for your continued engagement!  Please refer to our response to reviewer TSHw regarding the use of the same slice partition.

---

### Official Review · Reviewer_TSHw · 2024-07-14

**Soundness:** 3
**Presentation:** 2
**Contribution:** 3
**Rating:** 5
**Confidence:** 3

**Summary:**

This paper investigates why non-uniform bidding in first-price auctions causes inefficiency while uniform bidding does not. The authors propose a new model that partitions auctions into slices. Bidders are allowed to bid differently across different slices but need to bid uniformly in each slice. They characterize the price of anarchy of the game using the unbalancedness defined in the paper. The characterization shows that even just a partition of 2 slices will lead to inefficiency, and the balance across slices will improve the efficiency of the market.

**Strengths:**

The properties of the unbalancedness function and its relations with the PoA is interesting within the studied model. The fine-grained analysis between settings with the greatest bidding power and the least provides a reasonable explanation for the inefficiency caused by non-uniform bidding in first-price auctions.

**Weaknesses:**

1. The model assumption that all bidders have the same auction partitions is not compatible with the empirical results it claims to explain, as bidders may have different slices of auctions. While multi-channel bidding seems a reasonable application of the model, different users may participate in different sets of channels. Moreover, there is no central authority that can adjust the balancedness across channels, which limits practical applications of the insight.
2. The definition of unbalancedness lacks intuition.

**Questions:**

1. Can the results extend to cases where bidders have different slice partitions of auctions?
2. Can you provide more intuitions on the definitions of unbalancedness, or how it is derived?

**Limitations:**

None.

---

> ### Author Rebuttal · Authors · 2024-08-06
>
> Thank you for your insightful and detailed comments.
>
> **Bidders have different slice partitions**: first we'd like to note that our model is consistent with prior work on non-uniform bid scaling, and in particular, our results can be viewed as a theoretical explanation for the recent empirical study "Non-uniform Bid-scaling and Equilibria for Different Auctions: An Empirical Study" from Google Research -- they also consider environments where all bidders share the same partitions.  In light of this, we believe this model already captures the essence of non-uniform bidding in reality.  On the other hand, we do agree that a more general model with different partitions would be even better.  In such a model, one can probably extend our results in the following way:
> - Refine the definition of (un)balancedness.  In particular, the reasonable definition of balancedness of a slice should exclude bidders who don't participate in auctions of this slice.  In addition, the definition of unbalancedness should take into consideration the "adjacency" between bidders, where two bidders are adjacent if they are both active in some slice.  The resulting definition would probably be based on a "balancedness graph", which degenerates to our definition when the graph is complete.
> - Then, when establishing the bounds, one would follow the same high-level plan of forcing a worst-case equilibrium, but now under more constraints.  More specifically, one would probably need to match slices that "generate" surplus to those that "consume" surplus, subject to the constraint that the surplus of each bidder flowing out of a slice cannot exceed that bidder's market share.  One would need to find the worst-case matching that forces the worst welfare.
>
> One important consideration here is the tradeoff between the complexity of the model and the cleanliness / readability of the results.  The above plan for extending our results would probably result in a far more complicated bound that depends on more parameters of the market.  Given this, we'd like to argue that our approach achieves a reasonable balance on the spectrum.
>
> **Intuitions on unbalancedness**: the definition "naturally" arises in the process of trying to force a worst-case equilibrium.  The idea is that low welfare in auctions with autobidders is typically because in some auctions, competition is extremely low, so the winner gets surplus almost for free.  These bidders who have high surplus then spend the surplus in other auctions to compete with the "rightful winner", which ultimately hurts the welfare.  One observation here is that when the latter phenomenon happens, the surplus "burnt" to beat the rightful winner consists of two parts: the part that one overpays for auctions where they are the rightful winner, and the part one overpays to beat the rightful winner.  While the latter amount is always the same as the loss of welfare, the former does not contribute to this loss.  So, to cause loss more efficiently, one would like to minimize the former amount.  Fixing the total surplus burnt, this is done by burning more surplus on slices that are less balanced -- hence the definition of balancedness.
>
> Now, another observation is that the two phenomena can hardly happen simultaneously in one slice, because the first requires a low bid multiplier, and the second requires a high one.  So, to force the worst-case welfare, intuitively one would greedily allocate the most unbalanced slices to the second phenomenon and ensure that the rest of the slices generate just enough surplus for burning.  This corresponds to an intricate optimization problem over a set of functions, which turns out to be controlled by the balancedness of each slice roughly in the way that the unbalancedness is defined.

---

> > ### Comment · Reviewer_TSHw · 2024-08-10
> >
> > Thanks for the responses. I have no further questions about the intuitions on unbalancedness. However, as for the first response, the reason that the previous empirical paper considered the same setting is not convincing to me. A practical scenario that fits in the setting should be better. Also, even if extending to the setting where bidders have different slices can lead to complicated bounds, some robust analysis would help strengthen the results and setting.

---

> > > ### Author Response · Authors · 2024-08-13
> > >
> > > Thank you for your continued engagement!  See below for our further response.
> > >
> > > We note that slices can arise not only exogenously because of the existence of multiple channels or platforms, but also endogenously from the implementation of the autobidding algorithm. Since optimal bidding is computationally costly, one may view slice-based uniform bid-scaling as a fine-grained approximation to optimal bidding. That is, the (approximately optimal) bidding algorithm has an internal partition of all auctions into slices (e.g., based on features of the user, such as language, OS, region, etc.), and performs uniform bid-scaling on each slice. The finer this partition is, the closer the algorithm is to optimal bidding. Practically, autobidding algorithms are usually maintained by online platforms. For bidders using the same bidding product on the same platform, the same bidding algorithm is often used and it is natural to assume bidders have the same slices in such scenarios. Under this interpretation, our results suggest that "partially" optimal autobidding algorithms lead to worse auction efficiency.
> > >
> > > As for robust analysis under relaxed assumptions, as discussed in the original response, a fully general fine-grained bound would be quite cumbersome. Also, if we incorporate robustness by considering the worst case over arbitrary partitions, the bound would degrade to $1/2$.
> > >
> > > However, one can establish the following "in-between" statement: the PoA cannot be better than the one given in our Theorem 1 for any market A that can be divided into a set of submarkets B1 ... Bk, where not all bidders participate in each submarket, but in each submarket all participating bidders share the same slice partitions and participate in all slices. Here, balancedness is defined only on these bidders and our bound applies to each submarket individually.
> > >
> > > We can develop a PoA bound by establishing a "dominance" relationship between the market A and a weighted aggregation of submarkets B1 ... Bk, in which each slice in market A is no more balanced than the corresponding slice in the weighted sum. The resulting bound would be the sum of bounds for each Bi weighted in the same way. This enables a bound for markets in which all bidders share the same slice partitions but may not participate in all slices (corresponding to multi-channel scenarios). We believe it's also possible to derive "in-between" bounds for cases where bidders have different slice partitions by refining the above approach.

---

### Official Review · Reviewer_DAvE · 2024-07-14

**Soundness:** 3
**Presentation:** 3
**Contribution:** 3
**Rating:** 7
**Confidence:** 3

**Summary:**

This paper studies the efficiency, measured by the price of anarchy, of a multi-round single-item first-price auction involving multiple bidders. The auctions are segmented into multiple slices, within each of which all bidders utilize uniform bidding strategies. While the bidding parameters of each bidder must remain consistent within each slice, they can vary across different slices.

The authors define the unbalanceness of the multi-round auction based on all the valuations of the bidders. Based on such unbalanceness, they show that the price of anarchy of the multi-round auction is $1 - \frac{1}{2}t$ for $t \in [2/5, 1]$, where $t$ is the upper bound of the unbalanceness. As the unbalancedness increases with more subdivisions of the slices, the authors conclude that more sophisticated auto-bidders lead to less efficient auction outcomes.

**Strengths:**

1. The fine-grained slice-based model is a powerful tool for analyzing the equilibrium outcomes of more sophisticated autobidders compared to simple uniform-bidding strategies.
2. The theoretical results are insightful, characterizing the relationship between the price of anarchy and the sophistication level of autobidders, using unbalancedness as the key intermediary.

**Weaknesses:**

There remains a gap between realistic repeated first-price auctions and the fine-grained slice-based modeling. In reality, budget-constrained autobidders dynamically adapt their bidding strategies (such as through pacing) based on auction outcomes. However, in the paper's model, the authors restrict bidding strategies to static uniform bidding within each slice. Despite this, I still find the theoretical results valuable.

**Questions:**

None

**Limitations:**

See the weaknesses.

---

> ### Author Rebuttal · Authors · 2024-08-06
>
> Thank you for your insightful and encouraging comments.
>
> **Practicality of (per-slice) uniform bidding**: while autobidders in reality are presumably more sophisticated, there is evidence that the advertising industry finds per-slice uniform bidding a reasonable approximation. In particular, the recent paper "Non-uniform Bid-scaling and Equilibria for Different Auctions: An Empirical Study" from Google Research considers a form of non-uniform bidding strategies essentially identical to per-slice uniform bidding.  In their paper, they "define partitions to the queries" where "for each partition d, a non-uniform bid-scaling strategy chooses one bid multiplier", and study "how the simulation results vary with the granularity of the partition".  Our results can be viewed as a theoretical explanation of their empirical findings.  As such, we'd argue that conceptually, our results have similar practical implications.

---

### Decision · Program_Chairs · 2024-09-25

**Decision:**

Accept (poster)

**Comment:**

Reviewers more or less agreed that the general topic is interesting and the results appear to be non-trivial and mathematically interesting. On the negative side, all reviewers raised concerns on the practicality of the results due to possibly unnatural assumptions: static uniform bidding within each slice and all bidders have the same auction partitions. After the rebuttal/discussions, all reviewers remained unconvinced, yet the general consensus is that the interesting results outweigh the strong assumptions. After all, the paper is a solid and non-trivial piece of work. It may not be the last paper in this line of research but may inspire future work. That being said, having more discussions on the motivation and the assumptions would make the paper and future work stronger. (Generally, I think "it was assumed in a previous paper by X group" is not a very convincing argument). We hope the authors find the reviews helpful. Thanks for submitting to NeurIPS!